# Genuine Reversible Data Hiding Technique for H.264 Bitstream Using Multi-Dimensional Histogram Shifting Technology on QDCT Coefficients

**Jinwoo Kang, Hyunjung Kim and Sang-ug Kang \***

Department of Computer Science, Sangmyung University, Seoul 03016, Korea; 201411081@sangmyung.kr (J.K.); halena228@naver.com (H.K.)

**\*** Correspondence: sukang@smu.ac.kr; Tel.: +82-2-781-7588

**Abstract:** Video has become the most important medium for communication among people. Video has become the most important medium for communication among people. Therefore, reversible data hiding technologies for video have been developed so that information can be hidden in the video without damaging the original video in order to be used in the copyright protection and distribution field of video. This paper proposes a practical and genuine reversible data hiding method by using a multi-dimensional histogram shifting scheme on QDCT coefficients in the H.264/AVC bitstream. The proposed method defines the vacant histogram bins as a set of $n$-dimensional vectors and finds the optimal vector space, which gives the best performance, in a $4 \times 4$ QDCT block. In addition, the secret message is mapped to the optimal vector space, which is equivalent to embedding the information into the QDCT block. The simulation results show that the data hiding efficiency is the highest among the compared five existing methods. In addition, the image distortion and maximum payload capacity are measured quite high.

**Keywords:** reversible data hiding; H.264/AVC; histogram shifting, QDCT; genuine RDH; multi-dimensional histogram shifting

## 1. Introduction

With the fast and inexpensive network environment and the increasing distribution of digital content, there is growing concern about copyright infringement as well. One way to prevent copyright infringement is to hide copyright information such as copyright holders, camera source identification number, and distributors in the digital content, and then use the secretly hidden information as evidence when the content is illegally used later. Such copyright information may change from time to time for various reasons, so it must be deleted and rewritten again and again. However, information concealment using existing watermarking techniques can result in more or less damage to the original content. Thus, the copyright information is modified frequently, and the content quality is deteriorated accordingly. The reversible data hiding technique inserts and extracts data without compromising the original content. Therefore, if you use reversible data hiding technology to hide the copyright or distribution information in your content, there is no need to worry about the content damage caused by frequent information modifications. Moreover, since the copyright information tends to be large, which causes more damage to the content if a traditional watermarking technique is used, the reversible hiding technique is further required. The application fields are not limited to copyright protection. Images are used to monitor and visualize natural phenomena in various scientific research activities. Securing the originality and reliability of these images is also an important issue [1] and can be achieved by reversibly hiding time stamp or hash information.

## 2. Literature Review

Most reversible data hiding (RDH) algorithms are developed in either spatial or frequency domains. In the spatial domain, the difference expansion (DE) method proposed by Tian [2] and the histogram shifting (HS) method devised by Ni et al. [3] are known as two landmark works. The DE method doubles the difference between two neighboring pixel values and hides one secret bit in the least significant bit of the doubled difference. However, this method can cause fairly large distortions and requires a location map of data hiding positions. The HS method eliminated the need for the location map and significantly reduced distortion. The histogram of pixel values of a natural image has the peak bin and zero bin. A secret message is hidden in pixels corresponding to the peak bin and then all bins between the peak and zero bin are shifted toward the zero bin direction. The number of vacant bins created by shifting the histogram bins is proportional to the amount of data hiding, which is called payload, and distortion in the image. Hong et al. [4] improved the HS method in [3] by using a histogram of the prediction error of pixel value instead of pixel value itself. Both DE and HS methods are used in the frequency domain as well. Chang et al. [5] chose two successive zeros within the medium-frequency component region in an $8 \times 8$ quantized discrete cosine transform (QDCT) block as a place to hide data. In the JPEG image compression standard, QDCT coefficients are widely used data hiding places because QDCT coefficients are entropy-coded and then finally converted to bitstream without loss of information. Wang et al. [6] utilized the fact that a DCT value is equal to the scalar multiple of the corresponding QDCT value, where the scalar is determined by a quantization table. A specific position of the quantization table is divided by a constant k and then the corresponding QDCT value is multiplied by the same k to create k-array data hiding spaces. With this strategy, the image distortion is alleviated more or less.

Since video is an important and popular content format, RDH technology has been naturally applied to video contents, especially to H.264/Advanced Video Coding [7]. Liu et al. [8] embedded data only in $4 \times 4$ QDCT coefficients that were not used for intra prediction in order to avoid distortion propagation to other blocks and other frames. The mid and high frequency components in the selected blocks are used to embed data with the HS method. Gujjunoori et al. [9] embedded two bits into mid-frequency coefficients of an $8 \times 8$ QDCT block by considering the human visual system. The result showed that the payload is doubled compared to [5] with better subjective image quality, measured in the PSNR-HVS and PSNR-HVS-M metrics. Shaid et al. [10] suggested an approach similar to [8] to minimize error propagation due to the intra prediction mode. They found some solution patterns that do not affect the rightmost and the bottom seven pixels, which are used as predictors of the intra prediction. In addition, then, secret bits are hidden into nonzero positions in those solution patterns. However, those are located in the low frequency region instead of the medium frequency region, resulting in relatively high distortion in the stego video. Bouchama et al. [11] devised a mapping table that maps three bit long secret messages into four QDCT coefficient values in the mid-frequency region. With this approach, the data hiding capacity is increased, and PSNR is improved about 0.44 dB compared to the method in [9] that can hide two bits per $4 \times 4$ QDCT block. Kim et al. [12] hid the secret message in the same location as [11], but they was able to hide up to four bits per block and the number of hidable blocks increased compared to [11]. In addition, the distortion caused by data hiding is suppressed by the compensation effect that can happen in specific conditions. When the compensation occurs, there's no distortion and file size increase due to data hiding. In addition, a novel concept of genuine RDH is proposed because the previous RDH methods embed data during the compression process and extract data during the decompression process. With this approach, the original video cannot be fully recovered because of the lossy compression characteristics of the H.264 standard, which leads to the unintended result of pseudo reversibility. Chen et al. [13] modified the last QDCT coefficient to minimize the impact on visual quality. Three bits can be embedded into one pair of coefficients, consisting of two last coefficients from different QDCT blocks. This approach can yield a fairly good PSNR, but the payload is small because at most three bits can be hidden in two

blocks. Moreover, the size of the stego bitstream has been significantly increased due to the properties of the entropy coding algorithm.

Even though most reversible video data hiding technologies focus on high payload, low distortion, and small increase of compressed bitstream size, there are application-specific approaches. Ma et al. [14] applied the RDH technique to privacy protection of surveillance videos. The privacy region such as a person's face is visually protected with full reversibility so that the privacy is protected under normal circumstances and, at the same time, full information is available to law enforcement. As part of privacy protection, RDH methods in encrypted video have been conducted recently. In [15], specific code words substitution method is used for three important syntax elements including DCT coefficients of an encrypted H.264 video. Yao et al. [16] theoretically analyzed the picture distortion caused by data embedding and inter-frame distortion drift. Xu [17] proposed a way of hiding data in a partially encrypted HEVC video using the coefficient modulation method. In recent years, many techniques used in H.263 have also been applied to HEVC standard [18] in [19,20].

Most previous works are pseudo reversible and not practical. Since most videos are distributed in a compressed form, it is desirable to utilize H.264 bitstream as a cover video as in [12,14]. In this paper, we propose a genuine RDH technique that utilize a histogram shifting method on mid-frequency QDCT coefficients. The composition of this paper is as follows: Section 3 explains the proposed algorithm of reversible data hiding and extraction, Section 4 explains the experimental environment and results, and Section 5 discusses and presents conclusions on the findings of simulation and its implications.

## 3. The Proposed Method

The overall reversible data hiding and extraction process shown in Figure 1 adapted the genuine method proposed in [12]. First, the cover H.264 bitstream compressed by an H.264 encoder is decompressed until the QDCT coefficients are obtained. Second, some secret message is embedded in certain positions in the QDCT block so that some QDCT coefficients are modified causing image distortion. Finally, the modified blocks are re-compressed to make a stego H.264 bitstream. The data extraction process is reversed. The stego bitstream is the input to the extraction process, and the fully recovered H.264 bitstream and the extracted secret message are the output.

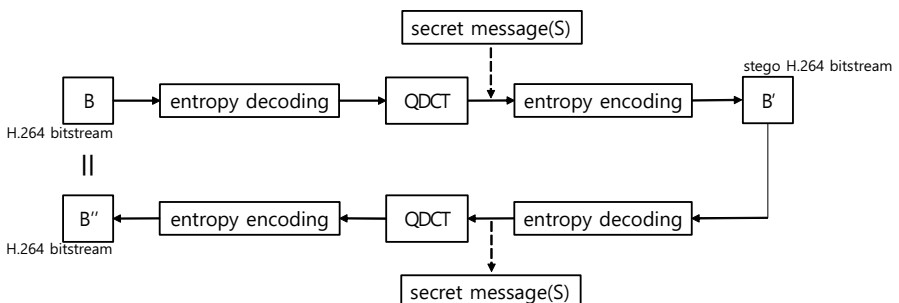

**Figure 1.** The overall reversible data hiding and extraction process.

### 3.1. Hidable QDCT Block Identification

Network Adaptation Layer units containing an I slice or P slice include a macroblock header and residual data, which are converted into $4 \times 4$ or $8 \times 8$ QDCT blocks during the decompression process. Only $4 \times 4$ blocks are used for data hiding, while $8 \times 8$ blocks are skipped. One $4 \times 4$ QDCT block has 16 transform coefficients as shown in Figure 2 and it can be represented as a set of 16 elements denoted by $R = \{r_i \mid i = 1, 2, \ldots, 16\}$. In addition, the secret message to be hidden in R also can be denoted by a set $S = \{s_i \mid i \in Z^+\}$. Since the information of $R$ corresponds to the residual image after performing the prediction and motion estimation processes, the influence on the image distortion due to the coefficient modification of R may not be large. However, recognizing the fact that the distortion of the current frame can easily propagate to the next frames according to the H.264 compression method, the coefficient modification should be minimized.

Identifying QDCT blocks that are able to hide the secret data are the first step in the proposed RDH technique. Thus, let us define an expandable block as $R_e = \{r_i \in R \mid r_{11\sim16} = 0\}$ in order to select QDCT blocks with less complex textual patterns in the spatial domain. $R_e$ is defined in this way because the human visual system is more sensitive to images with high frequency components than images with low frequency components. The expandable blocks are histogram shifted and some of them become hideable blocks $R_h$, in which the secret message is actually hidden. The definition of $R_h$ changes according to histogram shifting thresholds and data hiding locations. Let's assume that $m$ consecutive bits of the secret message are hidden in one single $R_h$. We need at least $2^m$ histogram vacant bins to accommodate $m$ bits. According to the result of [11,12], the most appropriate positions for the data hiding are $r_7$ through $r_{10}$, taking into account the three main objectives of a RDH technology: larger payload, less distortion, and smaller bitstream increase. In this medium frequency range, the results are generally not biased between the aforementioned three contradictory goals. Thus, the hideable block is defined as $R_h = \{r_i \in R_e \mid r_{7\sim10} = 0\}$ because $r_{7\sim10} = 0$ usually constitutes the peak bin of a 4-dimensional histogram with axes $r_7$ through $r_{10}$.

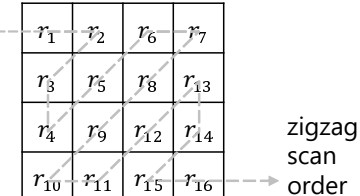

**Figure 2.** The representation of 4x4 QDCT coefficients.

### 3.2. Multi-Dimensional Histogram Shifting Method

The four-dimensional histogram is generated by counting the values of $r_7$ through $r_{10}$ of all $R_e$ blocks. The blocks corresponding to the peak bin are selected as $R_h$ in order to efficiently vacate adjacent bins to hide the secret message. The histogram is shifted toward zero bins along the four axes by the positive threshold $T_p$ in the positive direction and the negative threshold $T_n$ in the negative direction, respectively. As a result, the number of vacant bins $N_b = (T_p - T_n + 1)^4$ is proportional to both the number of chosen $r_i$s, which is fixed to 4 in this proposed method, and the threshold levels. Even though all the vacant bins can be used to hide secret data theoretically, some bins are not profitable because they may distort the stego image more than others. Therefore, a more sophisticated approach is needed to effectively hide the secret message in empty histogram bins.

There are several rules to hide the secret data into QDCT coefficients. First, $m$ consecutive bits of secret data are hidden in the first $R_h$ and the next $m$ consecutive bits in the next $R_h$. This rule helps to increase the hiding capacity according to the Pigeonhole principle because all $R_h$s with the same capacity are assigned the same amount of secret data. The selected $r_i$s used to hide the data should be consecutive in zigzag scan order as shown in Figure 2, and minimum numbers of them have nonzero values after data hiding. By doing so, the modified QDCT coefficients $r_i'$s are likely to have successive zero values, which is advantageous for making short entropy code length and less stego bitstream increase accordingly. In this paper, we chose $r_7$ to $r_{10}$ for data hiding and represented them as a vector $\vec{r} = (r_7, r_8, r_9, r_{10})$. If we define all possible values of $\vec{r}$ as the vector space $V$, the vacant bins are a subspace of $V$ and called vacant bin space $V_v$. The space $V_v$ depends on $(T_n, T_p)$ and is defined as in Equation (1):

$$V_v = \{\vec{r} \mid T_n \leq r_{7\sim10} \leq T_p\} \tag{1}$$

However, the only a portion of $V_v$ is used for data hiding in order to reduce image distortion and the stego bitstream size at the cost of payload reduction. Thus, an $m$-sized data chunk should be embedded in a coded fashion to achieve relatively good video quality. For example, if '1111' data are embedded in $R_h$ as it is, then $\vec{r} = (1, 1, 1, 1)$ is filled, causing a severe distortion by modifying

$\overrightarrow{r} = (0,0,0,0)$ to $\overrightarrow{r} = (1,1,1,1)$. However, if '1111' is embedded into $\overrightarrow{r} = (1,0,0,0)$ in a coded way, then the effect is relatively small. We also can define the subspace $V_u$ in $V$ as in Equation (2), which is used to hide data:

$$V_u = \{\overrightarrow{r} \mid \|\overrightarrow{r}\| \leq \max(|T_n|, |T_p|), \overrightarrow{r} = k\overrightarrow{r_{bi}}, T_n \leq k \leq T_p\} \tag{2}$$

where $k$ is integer and $\overrightarrow{r_{bi}}$ is standard basis vectors that span $V$. Since the dimension of $V$ is four, there are four $\overrightarrow{r_{bi}}$s: $\overrightarrow{r_{b1}}$, $\overrightarrow{r_{b2}}$, $\overrightarrow{r_{b3}}$ and $\overrightarrow{r_{b4}}$. The size of $V_u$, or $|V_u|$, is proportional to the embedding capacity of a single $R_h$ and can be calculated using Equation (3):

$$|V_u| = 4(T_p - T_n) + 1 \tag{3}$$

The smaller the norm of $\overrightarrow{r}$, or $\|\overrightarrow{r}\|$, the less image distortion. For example, if $(T_n, T_p) = (0,1)$, then $V_v = \{(0,0,0,0), (0,0,0,1), \ldots, (1,1,1,0), (1,1,1,1)\}$ and $|V_v| = N_b = (1 - 0 + 1)^4 = 16$. The set of used vacant histogram bins is $V_u = \{(0,0,0,0), (0,0,0,1), (0,0,1,0), (0,1,0,0), (1,0,0,0)\}$ and the number of the elements of the set is $|V_u| = 5$. If $(T_n, T_p) = (-1,2)$, then $|V_u| = 13$ and $V_u = \{(0,0,0,0), (0,0,0,1), (0,0,1,0), (0,1,0,0), (1,0,0,0), (0,0,0,-1), (0,0,-1,0), (0,-1,0,0), (-1,0,0,0), (0,0,0,2), (0,0,2,0), (0,2,0,0), (2,0,0,0)\}$. Since the secret message can be mapped to these used vacant bins, the payload capacity of $R_h$ is determined by Equation (4):

$$Cap(R_h) = \log_2 |V_u|, [bits] \tag{4}$$

For instance, if $(T_n, T_p) = (-1,1)$, then $Cap(R_h) = 3.17$ bits meaning that we can hide 3.17 bits per $R_h$. Thus, three bits are always hidden in a $R_h$ and sometimes four bits can be hidden as shown in Table 1. According to Table 1, the $R_h$ can hide four bits when the data chunk of $S$ contains '1110' or '1111'. The one-to-one correspondence table depends on $(T_n, T_p)$, so the specific mapping rule should be designed accordingly. The proposed HS method can cause QDCT coefficient overflow. Considering that the QP value ranges from 10 to 40 in practice, it is reasonable to assume that there are no overflow issues.

**Table 1.** An example of a one-to-one correspondence table from the data chunk of $S$ to $V_u$.

| S | $\overrightarrow{r}$ | S | $\overrightarrow{r}$ | S | $\overrightarrow{r}$ |
|---|---|---|---|---|---|
| 000 | (0,0,0,0) | 011 | (0,1,0,0) | 110 | (0,0,−1,0) |
| 001 | (0,0,0,1) | 100 | (1,0,0,0) | 1110 | (0,−1,0,0) |
| 010 | (0,0,1,0) | 101 | (0,0,0,−1) | 1111 | (−1,0,0,0) |

*3.3. Reversible Data Hiding and Extraction Algorithms*

　　For a better understanding, the overall data embedding and extraction algorithms are summarized in the following Algorithms 1 and 2.

---

**Algorithm 1:** Data Embedding Algorithm and Procedure.

---

　**Input:** H.264/AVC compressed cover bitstream $B$, secret message $S$ and threshold $(T_n, T_p)$.
　**Output:** H.264/AVC compressed stego bitstream $B'$.
　**Step 1** Calculate the used vacant bin space $V_u$ using Equation (2).
　**Step 2** Prepare a one-to-one correspondence table from the data chunk of $S$ to $V_u$.
　**Step 3** Decode $B$ and find the first QDCT block $R$.
　**Step 4** Restore coefficient values of the $R$.
　**Step 5** Determine if the $R$ is a expandable block $R_e$.
　**Step 5-1** If yes, the $\overrightarrow{r}$ of $R_e$ is shifted by $(T_n, T_p)$ to make vacant histogram bins.
　**Step 6** Determine if the $R_e$ can be classified as $R_h$.
　**Step 6-1** If yes, map the data chunk of $S$ to corresponding element of $V_u$.
　**Step 6-2** If yes, replace the $\overrightarrow{r}$ of $R_h$ with the mapped element.
　**Step 7** Restore coefficient values of the next $R$ from the.
　**Step 8** Go to step 5 until all $R$ blocks are processed.

---

---

**Algorithm 2:** Data Extraction Algorithm and Procedure.

---

　**Input:** H.264/AVC compressed stego bitstream $B'$ and threshold $(T_n, T_p)$.
　**Output:** H.264/AVC compressed restored bitstream $B$ and secret message $S$.
　**Step 1** Calculate the used vacant bin space $V_u$ using Equation (2).
　**Step 2** Prepare a one-to-one correspondence table from $V_u$ to the data chunk of $S$.
　**Step 3** Decode $B'$ and find the first QDCT block $R$.
　**Step 4** Restore coefficient values of the $R$.
　**Step 5** Determine if the $R$ is a hidable block $R_h$.
　**Step 5-1** If yes, extract the secret message $S$ using the $\overrightarrow{r}$ of $R_h$ and the mapping table prepared
　　in Step 2.
　**Step 5-2** If yes, set the $\overrightarrow{r}$ of $R_h$ to the zero vector.
　**Step 6** Determine if the $R$ is a hidable block $R_h$.
　**Step 6-1** If yes, the $\overrightarrow{r}$ of $R_h$ is shifted backward by $(T_n, T_p)$ to remove vacant histogram bins.
　**Step 7** Restore coefficient values of the next $R$ from $B'$.
　**Step 8** Go to Step 5 until all $R$ blocks are processed.

---

## 4. Simulation Results

　　We implemented the proposed method based on the H.264/AVC JM-18 reference software [21]. Totally, eight $352 \times 288$-sized video sequences of 300 frames were used for the simulation, including bridges(closed), Cost Guard, Foreman, Hall Monitor, Mobile, Mother, News, and Akiyo. JM-18 software configuration parameters were set to the baseline profile, 30 frames/second, and intra update period of 30 frames (i.e., group of frame IPPP...) with both CAVLC entropy and RD optimization mode on. As an indicator of video quality distortion, we used the peak signal-to-noise ratio (PSNR) between the compressed video $B'$ and the stego video $B''$. The payload per frame (PPF) index was used to measure the payload capacity of the algorithm. PPF is calculated by dividing the amount of hidden payload by the number of frames in the cover video. If the PPF value is large, it means that the payload performance is good. Meanwhile, the file increase per payload (FPP) index is used to measure the file growth effect after data embedding. FPP is the difference in file size before and after data hiding divided by the amount of hidden payload. Smaller FPP means good data hiding efficiency.

### 4.1. Searching for the Optimal Subspace $V_u$

There exist many $V_u$ spaces because $V_u$ can vary depending on relevant parameters as described in Equation (2). Thus, it is necessary to compare the performance of various $V_u$s to find out the optimal one. First, the difference between symmetric and asymmetric histogram shifting is investigated as shown in Table 2. Four asymmetric threshold cases $(T_n, T_p) = (0, 1), (-1, 0), (-1, 2), (-2, 1)$ are tested to measure the three main performances of the proposed method. We appended the letter $S$ to $(T_n, T_p)$ in order to indicate that the histogram is symmetrically shifted, even though the threshold is set to asymmetric. Therefore, the $\overrightarrow{r}$ of $R_e$ is not respectively shifted by $|T_p|$ in the positive direction and by $|T_n|$ in the negative direction, but shifted equally by $\max(|T_n|, |T_p|)$ in both directions. However, the $V_u$ of $(T_n, T_p)S$ is the same as $(T_n, T_p)$ according to Equation (2). For example, $\overrightarrow{r}$ is shifted by 1 equally in the positive and negative directions for $(T_n, T_p) = (-1, 0)S$ and by 1 in the negative direction only for $(T_n, T_p) = (-1, 0)$ while their used vacant bin spaces $V_u = \{(1, 0, 0, 0), (0, 0, 0, -1), (0, 0, -1, 0), (0, -1, 0, 0), (-1, 0, 0, 0)\}$ are the same. The PPF is set to the same amount to clearly compare the PSNR and FPP performances. All the nine sequences are simulated and the average values are written in Table 2. The symmetric shifting always shows better performance in PSNR than the asymmetric one. However, the results of FPP are reverse. The interim result from Table 2 is that the symmetric shifting method gives better image quality, and poorer file size increases at the same amount of payload. This is because natural image quality deteriorates significantly when asymmetric frequency components are introduced synthetically. Thus, we will adapt the symmetric shifting to achieve a higher PSNR. From now on, we will compare the performance differences for various $V_u$s as depicted in Table 3. We also investigated four cases of histogram shifting thresholds for all test sequences. Specifically, four cases $(T_n, T_p) = (-1, 0)S, (-1, 1), (-2, 1)S, (-2, 2)$ were tested, and the average values are recorded. We intentionally made the payload the same for the same QP value to tell which $V_u$ provides better PSNR and FPP performance. It is quite clear that the $(T_n, T_p) = (-1, 1)$ case reports the best PSNR and FPP at the same time, and we will use the corresponding $V_u$ as the optimal space.

**Table 2.** The performance comparison between symmetric and asymmetric histogram shifting.

| $(T_n, T_p)$ | | PSNR (dB) | | | | PPF (bits/frame) | | | | FPP | | | |
|---|---|---|---|---|---|---|---|---|---|---|---|---|---|
| | | 28 | 26 | 24 | 22 | 28 | 26 | 24 | 22 | 28 | 26 | 24 | 22 |
| (0,1) | $S^\dagger$ | 20.87 | 21.96 | 23.08 | 27.02 | 2880 | 3165 | 3705 | 4301 | 2.8 | 2.9 | 2.9 | 3.0 |
| | $A^\ddagger$ | 19.56 | 20.32 | 21.19 | 24.45 | 2880 | 3165 | 3705 | 4301 | 2.5 | 2.6 | 2.6 | 2.6 |
| (−1,0) | S | 19.64 | 21.56 | 23.53 | 24.37 | 2880 | 3165 | 3705 | 4301 | 2.8 | 2.9 | 2.9 | 3.0 |
| | A | 18.41 | 20.00 | 21.55 | 22.31 | 2880 | 3165 | 3705 | 4301 | 2.6 | 2.6 | 2.7 | 2.7 |
| (−1,2) | S | 22.75 | 23.38 | 24.44 | 28.00 | 4636 | 5094 | 5964 | 6924 | 2.4 | 2.5 | 2.5 | 2.6 |
| | A | 21.28 | 21.91 | 22.81 | 26.22 | 4636 | 5094 | 5964 | 6924 | 2.4 | 2.4 | 2.4 | 2.5 |
| (−2,1) | S | 21.28 | 23.74 | 25.93 | 26.36 | 4636 | 5094 | 5964 | 6924 | 2.5 | 2.6 | 2.6 | 2.7 |
| | A | 20.03 | 22.15 | 23.88 | 24.20 | 4636 | 5094 | 5964 | 6924 | 2.4 | 2.5 | 2.5 | 2.6 |

$S^\dagger$: Symmetric shifting, $A^\ddagger$: Asymmetric shifting.

**Table 3.** The performance comparison between various $V_u$s.

| QP | PSNR (dB) | | | | PPF (bits/frame) | | | | FPP | | | |
|---|---|---|---|---|---|---|---|---|---|---|---|---|
| | ① | ② | ③ | ④ | ① | ② | ③ | ④ | ① | ② | ③ | ④ |
| 28 | 22.59 | 32.95 | 26.11 | 29.72 | 846 | 846 | 846 | 846 | 2.79 | 2.29 | 2.46 | 2.50 |
| 26 | 24.25 | 34.83 | 27.82 | 31.11 | 1074 | 1074 | 1074 | 1074 | 2.90 | 2.35 | 2.56 | 2.58 |
| 24 | 26.09 | 35.04 | 29.88 | 31.68 | 1351 | 1351 | 1351 | 1351 | 2.92 | 2.37 | 2.58 | 2.61 |
| 12 | 26.55 | 36.57 | 29.93 | 33.70 | 1610 | 1610 | 1610 | 1610 | 3.02 | 2.44 | 2.71 | 2.72 |

①: $(T_n, T_p) = (-1, 0)S$, ②: $(T_n, T_p) = (-1, 1)$, ③: $(T_n, T_p) = (-2, 1)S$, ④: $(T_n, T_p) = (-2, 2)$.

### 4.2. Performance Comparison with Existing Methods

The performance of the proposed method is measured for PSNR, PFF, and FPP as before. The simulation results of the proposed method are compared with those of [10–13] as shown in Table 4. The performance of [10,11,13] is much lower than the proposed method just because the pseudo RDH method is used, apart from their own algorithms. For fair comparison, the genuine RDH method suggested in [12] is applied to those methods. In Figure 3, the tenth stego video frames of the 'Hall monitor' sequence at $QP = 30$ are displayed to compare image quality. From a perspective of image distortion, the overall comparison is illustrated in Figure 4a and Chen et al.'s method [13] achieves the highest average PSNR of 36.41 dB as we expected. The second highest is the proposed one at 29.76 dB. From the viewpoint of capacity, Kim et al.'s method [12] can hide the largest payload at 5974 bits per frame and the second highest is the proposed one at 4273 as shown in Figure 4b. On the other hand, the PPF of [13] is 1924, which is the smallest among the five methods and less than half of the proposed methods. Finally, the file size increase due to a data embedment is a very important performance factor as it implies how compatible the algorithm is with the H.264 standard. From the graph in Figure 4c, the proposed method shows the lowest ratio at 2.30, which is much better than 2.72 and 2.95 achieved by [12,13], respectively. Therefore, the proposed method is 18.3% more efficient than other methods for most test sequences.

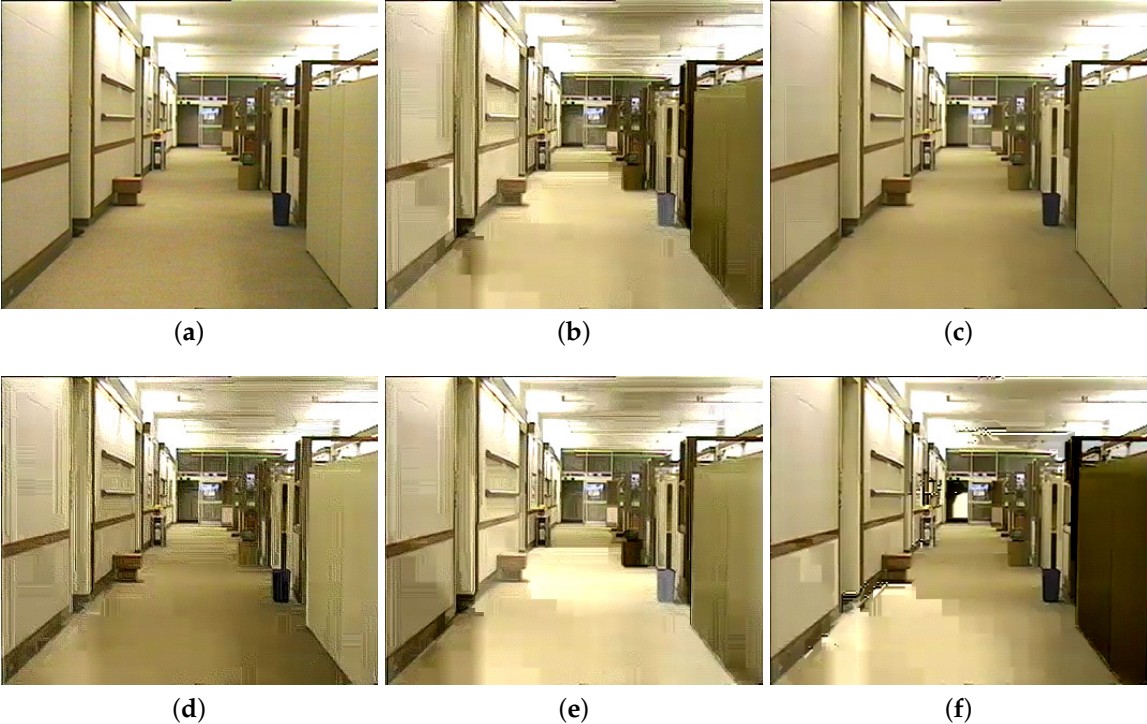

**Figure 3.** Tenth frame of the Hall monitor video sequence encoded with QP = 30 after data hiding up to full payload. (**a**) The original video; (**b**) The proposed stego video; (**c**) The stego video of [13]; (**d**) The stego video of [12]; (**e**) The stego video of [11]; (**f**) The stego video of [10].

**Table 4.** Comparison between our proposed method(P) and the methods of [10–13].

| Video | Performance | Method | QP | | | | Average |
|---|---|---|---|---|---|---|---|
| | | | 30 | 28 | 26 | 24 | |
| Bridge (close) | PSNR (dB) | P | 28.63 | 30.75 | 30.92 | 32.39 | 30.67 |
| | | [13] | 34.16 | 35.16 | 35.06 | 36.21 | 35.15 |
| | | [12] | 27.90 | 29.54 | 31.86 | 31.08 | 30.10 |
| | | [11] | 25.83 | 27.23 | 29.68 | 32.05 | 28.70 |
| | | [10] | 22.97 | 23.36 | 24.08 | 22.61 | 23.25 |
| | PPF | P | 1801 | 1812 | 1891 | 4231 | 2434 |
| | (bits/frame) | [13] | 1033 | 1368 | 1707 | 2671 | 1695 |
| | | [12] | 2675 | 2809 | 3006 | 6130 | 3655 |
| | | [11] | 1729 | 1739 | 1815 | 4060 | 2335 |
| | | [10] | 1241 | 1655 | 2073 | 3102 | 2018 |
| | FPP | P | 2.49 | 2.60 | 2.68 | 2.47 | 2.56 |
| | | [13] | 2.27 | 2.16 | 2.12 | 2.26 | 2.20 |
| | | [12] | 2.97 | 2.96 | 2.96 | 2.98 | 2.97 |
| | | [11] | 2.61 | 2.72 | 2.80 | 2.59 | 2.68 |
| | | [10] | 2.36 | 2.28 | 2.25 | 2.44 | 2.33 |
| Coastguard | PSNR (dB) | P | 24.17 | 27.07 | 29.14 | 31.01 | 27.85 |
| | | [13] | 33.47 | 35.66 | 35.82 | 38.43 | 35.84 |
| | | [12] | 25.02 | 25.88 | 28.17 | 30.49 | 27.39 |
| | | [11] | 23.26 | 23.16 | 26.66 | 29.38 | 25.61 |
| | | [10] | 19.03 | 19.24 | 20.38 | 20.52 | 19.79 |
| | PPF | P | 8725 | 9158 | 9137 | 8785 | 8951 |
| | (bits/frame) | [13] | 3185 | 4184 | 5125 | 6076 | 4642 |
| | | [12] | 11,672 | 13,055 | 13,934 | 14,133 | 13,199 |
| | | [11] | 8372 | 8787 | 8767 | 8429 | 8589 |
| | | [10] | 4384 | 5616 | 6656 | 7530 | 6047 |
| | FPP | P | 2.12 | 2.24 | 2.41 | 2.54 | 2.33 |
| | | [13] | 2.82 | 2.59 | 2.39 | 2.22 | 2.50 |
| | | [12] | 2.89 | 2.89 | 2.90 | 2.91 | 2.90 |
| | | [11] | 2.23 | 2.35 | 2.52 | 2.65 | 2.44 |
| | | [10] | 3.05 | 2.97 | 2.93 | 2.88 | 2.96 |
| Foreman | PSNR (dB) | P | 26.12 | 27.99 | 30.49 | 30.86 | 28.87 |
| | | [13] | 33.50 | 35.98 | 36.20 | 38.71 | 36.10 |
| | | [12] | 27.17 | 26.76 | 30.22 | 31.13 | 28.82 |
| | | [11] | 24.20 | 24.05 | 28.18 | 30.90 | 26.83 |
| | | [10] | 22.41 | 23.40 | 24.51 | 24.67 | 23.74 |
| | PPF | P | 4246 | 5409 | 6519 | 7716 | 5972 |
| | (bits/frame) | [13] | 1051 | 1521 | 2069 | 2794 | 1859 |
| | | [12] | 5379 | 6961 | 8548 | 10,229 | 7779 |
| | | [11] | 4074 | 5190 | 6255 | 7404 | 5731 |
| | | [10] | 732 | 1139 | 1668 | 2378 | 1479 |
| | FPP | P | 2.23 | 2.25 | 2.27 | 2.27 | 2.25 |
| | | [13] | 3.15 | 2.99 | 2.84 | 2.73 | 2.93 |
| | | [12] | 3.03 | 3.01 | 3.00 | 2.98 | 3.01 |
| | | [11] | 2.36 | 2.37 | 2.39 | 2.39 | 2.38 |
| | | [10] | 2.97 | 2.93 | 2.87 | 2.80 | 2.89 |
| Hall monitor | PSNR (dB) | P | 25.88 | 28.97 | 29.55 | 30.45 | 28.71 |
| | | [13] | 34.84 | 35.70 | 36.14 | 37.85 | 36.13 |
| | | [12] | 27.16 | 26.79 | 28.84 | 29.59 | 28.09 |
| | | [11] | 25.28 | 25.21 | 28.65 | 30.15 | 27.32 |
| | | [10] | 20.35 | 21.29 | 22.34 | 21.80 | 21.44 |

**Table 4.** *Cont.*

| Video | Performance | Method | QP | | | | Average |
|---|---|---|---|---|---|---|---|
| | | | 30 | 28 | 26 | 24 | |
| | PPF (bits/frame) | P | 1735 | 2522 | 3678 | 5447 | 3346 |
| | | [13] | 492 | 749 | 1151 | 1852 | 1061 |
| | | [12] | 2246 | 3293 | 4856 | 7235 | 4408 |
| | | [11] | 1664 | 2420 | 3529 | 5227 | 3210 |
| | | [10] | 374 | 596 | 983 | 1704 | 914 |
| | FPP | P | 2.18 | 2.19 | 2.22 | 2.22 | 2.20 |
| | | [13] | 2.87 | 2.82 | 2.76 | 2.72 | 2.79 |
| | | [12] | 2.93 | 2.93 | 2.93 | 2.92 | 2.93 |
| | | [11] | 2.28 | 2.29 | 2.32 | 2.31 | 2.30 |
| | | [10] | 3.42 | 3.30 | 3.18 | 3.05 | 3.24 |
| Mobile | PSNR (dB) | P | 28.98 | 30.35 | 32.39 | 35.01 | 31.68 |
| | | [13] | 32.75 | 34.35 | 35.19 | 37.19 | 34.87 |
| | | [12] | 26.91 | 28.69 | 31.48 | 32.95 | 30.01 |
| | | [11] | 26.53 | 28.65 | 30.92 | 33.82 | 29.98 |
| | | [10] | 22.16 | 21.81 | 23.77 | 23.72 | 22.87 |
| | PPF (bits/frame) | P | 7810 | 7793 | 7373 | 6855 | 7458 |
| | | [13] | 3145 | 4108 | 4919 | 5666 | 4459 |
| | | [12] | 10,893 | 11,416 | 11,286 | 10,677 | 11,068 |
| | | [11] | 7494 | 7478 | 7074 | 6577 | 7156 |
| | | [10] | 3316 | 4372 | 5239 | 5992 | 4730 |
| | FPP | P | 2.35 | 2.45 | 2.56 | 2.59 | 2.49 |
| | | [13] | 2.32 | 2.16 | 2.04 | 1.98 | 2.13 |
| | | [12] | 2.93 | 2.91 | 2.90 | 2.87 | 2.90 |
| | | [11] | 2.45 | 2.54 | 2.65 | 2.67 | 2.58 |
| | | [10] | 2.92 | 2.81 | 2.70 | 2.60 | 2.76 |
| Mother | PSNR (dB) | P | 28.10 | 28.40 | 29.64 | 30.65 | 29.20 |
| | | [13] | 36.83 | 37.46 | 38.09 | 39.37 | 37.94 |
| | | [12] | 29.25 | 28.82 | 31.99 | 33.25 | 30.83 |
| | | [11] | 26.80 | 26.46 | 29.49 | 31.07 | 28.46 |
| | | [10] | 26.63 | 25.95 | 26.12 | 26.03 | 26.18 |
| | PPF (bits/frame) | P | 1618 | 2174 | 2817 | 3652 | 2565 |
| | | [13] | 335 | 481 | 658 | 911 | 596 |
| | | [12] | 1966 | 2661 | 3481 | 4547 | 3164 |
| | | [11] | 1553 | 2086 | 2703 | 3504 | 2461 |
| | | [10] | 195 | 321 | 482 | 689 | 422 |
| | FPP | P | 2.15 | 2.13 | 2.13 | 2.14 | 2.14 |
| | | [13] | 3.63 | 3.50 | 3.39 | 3.32 | 3.46 |
| | | [12] | 3.03 | 3.01 | 2.99 | 2.98 | 3.01 |
| | | [11] | 2.27 | 2.26 | 2.25 | 2.26 | 2.26 |
| | | [10] | 3.00 | 2.93 | 2.89 | 2.84 | 2.92 |
| News | PSNR (dB) | P | 28.12 | 29.82 | 30.76 | 31.43 | 30.03 |
| | | [13] | 35.93 | 36.27 | 37.70 | 39.22 | 37.28 |
| | | [12] | 28.21 | 28.01 | 30.98 | 32.87 | 30.02 |
| | | [11] | 25.79 | 26.61 | 28.78 | 31.57 | 28.19 |
| | | [10] | 25.14 | 22.98 | 24.48 | 25.79 | 24.60 |
| | PPF (bits/frame) | P | 1639 | 1953 | 2258 | 2602 | 2113 |
| | | [13] | 477 | 622 | 780 | 982 | 715 |
| | | [12] | 2113 | 2559 | 3000 | 3487 | 2790 |
| | | [11] | 1573 | 1874 | 2167 | 2497 | 2027 |
| | | [10] | 408 | 547 | 701 | 889 | 636 |

**Table 4.** *Cont.*

| Video | Performance | Method | QP | | | | Average |
|---|---|---|---|---|---|---|---|
| | | | **30** | **28** | **26** | **24** | |
| | FPP | P | 2.17 | 2.20 | 2.24 | 2.24 | 2.21 |
| | | [13] | 2.90 | 2.77 | 2.72 | 2.64 | 2.76 |
| | | [12] | 2.93 | 2.93 | 2.94 | 2.92 | 2.93 |
| | | [11] | 2.28 | 2.31 | 2.34 | 2.34 | 2.32 |
| | | [10] | 3.55 | 3.45 | 3.36 | 3.27 | 3.41 |
| Akiyo | PSNR (dB) | P | 30.18 | 28.91 | 33.54 | 31.76 | 31.10 |
| | | [13] | 35.37 | 38.79 | 40.02 | 37.81 | 38.00 |
| | | [12] | 29.69 | 28.85 | 32.76 | 33.10 | 31.10 |
| | | [11] | 27.42 | 26.65 | 30.70 | 31.35 | 29.03 |
| | | [10] | 24.11 | 25.10 | 26.56 | 28.05 | 25.96 |
| | PPF (bits/frame) | P | 847 | 1175 | 1491 | 1877 | 1348 |
| | | [13] | 197 | 291 | 404 | 559 | 362 |
| | | [12] | 1070 | 1497 | 1922 | 2441 | 1732 |
| | | [11] | 813 | 1128 | 1431 | 1801 | 1293 |
| | | [10] | 145 | 225 | 330 | 470 | 293 |
| | FPP | P | 2.19 | 2.21 | 2.23 | 2.23 | 2.21 |
| | | [13] | 3.07 | 3.00 | 2.93 | 2.86 | 2.97 |
| | | [12] | 3.00 | 2.99 | 2.99 | 2.97 | 2.99 |
| | | [11] | 2.31 | 2.32 | 2.33 | 2.33 | 2.33 |
| | | [10] | 3.92 | 3.92 | 3.98 | 3.93 | 3.94 |

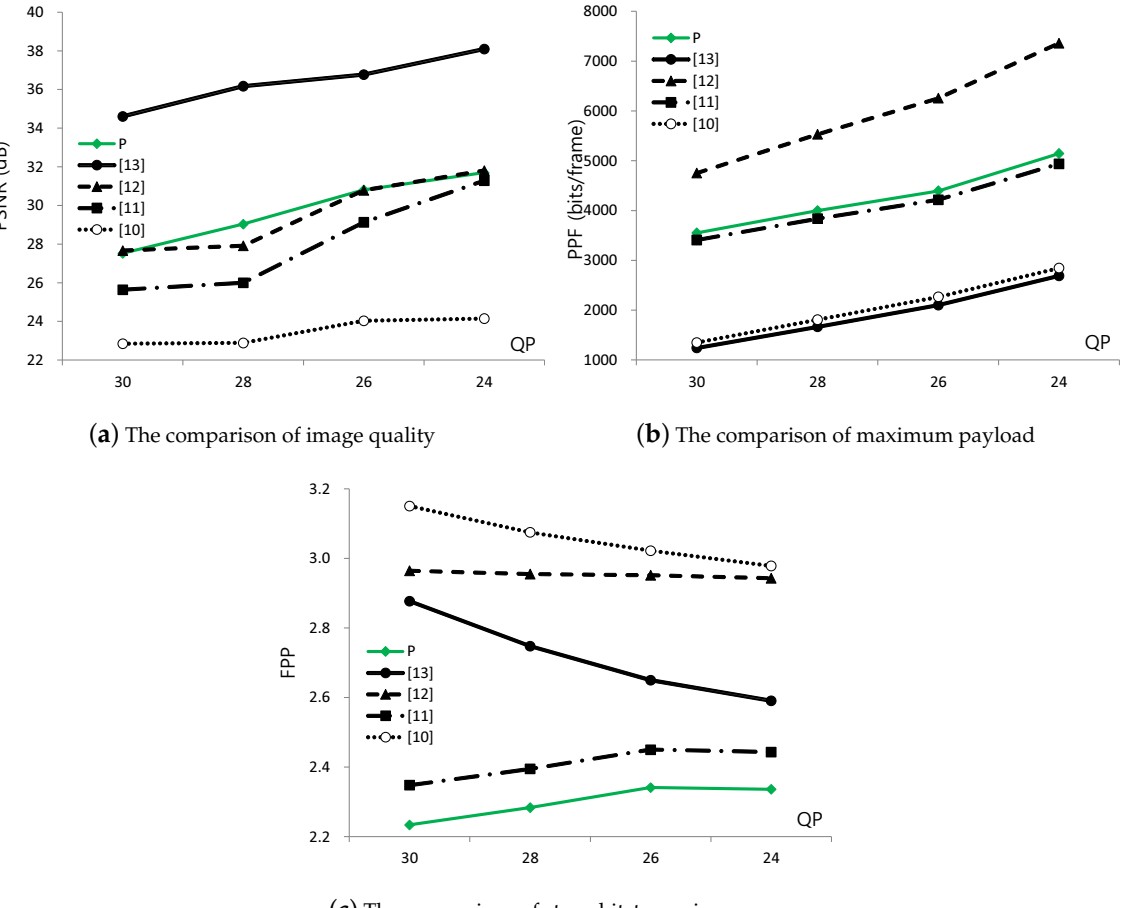

(**a**) The comparison of image quality

(**b**) The comparison of maximum payload

(**c**) The comparison of stego bitstream increase

**Figure 4.** Comparison between our proposed method (P) and the methods of [10–13].

## 5. Discussion

There are three main contributions of the paper. First, we proposed the generalized multi-dimensional HS method for H.264 bitstream. It should be also noticed that the method by [11] can be considered as a special case of the proposed method. As a result of generalization, it is possible to flexibly change the data embedding capacity by changing $(T_n, T_p)$ and $V_u$ as needed. On top of the flexibility, we can estimate the PSNR, PFF, and FPP by calculating the norm of $V_u$ elements and the size of $V_u$ that is nice to design the subspace $V_u$. Second, we found an optimal $V_u$ through a number of simulations. However, finding the optimal $V_u$ is closely related to performance, so there is still room for improvement if you devise a more sophisticated algorithm. Third, the proposed algorithm achieved best data hiding efficiency while maintaining quite good image quality and maximum payload capacity. The method by Chen et al. [13] shows good image quality but embeds the smallest amount of payload among the compared methods. The method by Kim et al. [12] hides the largest amount of secret messages, but the image quality is moderate and coding efficiency is on the lower side. Overall, the proposed method gave the best results in one of the three performances measured, and the top in two. Therefore, it can be said that the proposed method is the best. In fact, considering the situation in which RDH technology has evolved considerably, it is not easy to develop an algorithm with the best performance in all fields. Therefore, it is necessary for future research to define an application field first, and then develop a method suitable for it.

**Author Contributions:** Conceptualization, S.-u.K.; Data curation, J.K.; Formal analysis, H.K.; Investigation, J.K.; Methodology, S.-u.K.; Project administration, S.-u.K.; Software, J.K. and H.K.; Writing—original draft, S.-u.K. All authors have read and agreed to the published version of the manuscript.

**Funding:** This work was supported by the National Research Foundation of Korea grant funded by Korea government (NRF-2018R1A2B6006754).

**Conflicts of Interest:** The authors declare no conflict of interest. The funders had no role in the design of the study; in the collection, analyses, or interpretation of data; in the writing of the manuscript, or in the decision to publish the results.

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
