# Peer review of "Genuine Reversible Data Hiding Technique for H.264 Bitstream Using Multi-Dimensional Histogram Shifting Technology on QDCT Coefficients"

_applsci, doi:10.3390/app10186410_

Round 1
Reviewer 1 Report
The authors widely prove that the method is suitable. In my opinion the obtained results are very good. The simulation results validate the algorithm. Even if the approach does appear very simple its effectiveness does appear very appealing.
Indeed the algorithm could discover information derived by measurement images not only referred to secrete information. I strongly recommend that the approach is general and looking at various advanced application I remark that in some fields where image processing are fundamental to achieve the correct information
like in innovative device characterization the suitable reformulation of the algorithm is appropriate. I therefore suggest to include the following paper:
BiomicrofluidicsVolume 4, Issue 2, June 2010, Article number 014002BMF
Inizio modulo
Fine modulo
A polymeric micro-optical interface for flow monitoring in biomicrofluidics(Article)
- Sapuppo, F.aEmail Author,
- Llobera, A.b,
- Schembri, F.a,
- Intaglietta, M.c,
- Cadarso, V.J.b,
- Bucolo, M.a
This remark make the paper more appealind to a wider range of readers and open new application fields to such type of the algorithm.
It is a good paper congratulations to the authors.
Author Response
Thank you for your positive comments.
We added the following statement in the Introduction part. I hope this modification reflects your valuable comment.
Page 1>> The application fields are not limited to copyright protection. Images are used to monitor and visualize natural phenomena in various scientific research activities. Securing the originality and reliability of these images is also an important issue [1] and can be achieved by reversibly hiding time stamp or hash information.
Reviewer 2 Report
The paper follows the same structure as the previous publication https://link.springer.com/article/10.1007/s11042-017-4698-6, however shows improved algorithm. The paper is original, well written and clearly presented. valuable is the comparison of different methods of data hiding techniques, and the clear statement of the genuine contribution of the authors.
The topic itself attract little interest and has low potential of attraction of readers.
Author Response
Thank you for your positive comments. Even though you made little comment on this paper, we improved it by taking other reviewer’s valuable comments and modifying unclear statements.
Reviewer 3 Report
The reported work is of research significance, and the proposed method is novel to some extent. Overall, the provided experiments seem to prove better performances of the proposed algorithm compared to other methods. However, there are some issues the authors should address in order to make their work publishable.
- The introduction should be split into two parts: Introduction and Literature review.
- Section 2 is hard to read. More explanations are needed. For instance, explain the role of Tp and Tn. Which are the most commonly used values of the parameters? How does the performance of the algorithm depend on these parameters? Explain the notation used to define Re – r11~16
- In Section 3 – define the performance measures PSNR, PPF and FPP. Also, for each measure, you should indicate which class of values corresponds to good results. For instance, in case the value of PSNR is large, the inputs are similar.
Author Response
- The introduction should be split into two parts: Introduction and Literature review.
This is a great point. We divided the introductory part into the introductory part and the literary review part.
- Section 2 is hard to read. More explanations are needed. For instance, explain the role of Tp and Tn. Which are the most commonly used values of the parameters? How does the performance of the algorithm depend on these parameters? Explain the notation used to define Re – r11~16.
Thank you for pointing out this ambiguity in this paper. We have to confess that we made a mistake in expressing this parameter. (Tn, Tp) is the right order instead of (Tp, Tn). We corrected the all mistake made throughout the paper. Sorry for troubling you and thank you again. Also, we made our best to explain the role of every parameter shown in the paper.
Page 4>> Re is defined in this way because the human visual system is more sensitive to images with high frequency components than images with low frequency components.
- In Section 3 – define the performance measures PSNR, PPF and FPP. Also, for each measure, you should indicate which class of values corresponds to good results. For instance, in case the value of PSNR is large, the inputs are similar.
Thank you for your comment. We modified as following.
Page 6>> The payload per frame (PPF) index was used to measure the payload capacity of the algorithm. PPF is calculated by dividing the amount of hidden payload by the number of frames in the cover video. If the PPF value is large, it means that the payload performance is good. Meanwhile, file increase per payload (FPP) index is used to measure the file growth effect after data embedding. FPP is the difference in file size before and after data hiding divided by the amount of hidden payload. Smaller FPP means good data hiding efficiency.